# Risk Factors for Orbital Implant Extrusion after Evisceration

**DOI:** 10.3390/jcm10153329

**Published:** 2021-07-28

**Authors:** Ju-Mi Kim, Jae-Yun Sung, Hyung-Bin Lim, Eun-Jung Choi, Sung-Bok Lee

**Affiliations:** 1Department of Ophthalmology, Chungnam National University College of Medicine, Daejeon 35015, Korea; jjukkum2@naver.com (J.-M.K.); ssungjy@naver.com (J.-Y.S.); cromfans@hanmail.net (H.-B.L.); d19020k0@cnuh.co.kr (E.-J.C.); 2Department of Ophthalmology, Chungnam National University Sejong Hospital, Sejong 30099, Korea

**Keywords:** evisceration, extrusion, endophthalmitis, endogenous endophthalmitis, cellulitis, risk factor

## Abstract

This study analyzed risk factors for extrusion of orbital implants after evisceration by comparing patients with and without implant extrusion. Methods: We retrospectively reviewed the medical records of patients who underwent evisceration with primary implant placement by a single surgeon from January 2005 to December 2019 at the Chungnam National University Hospital. Age, sex, underlying systemic diseases, axial length of the fellow eye, the cause of evisceration, endophthalmitis type, implant type and size, and preoperative computed tomography findings were evaluated. Logistic regression analysis was used to identify the risk factors for implant extrusion. Results: Of the 140 eyes of 140 patients, extrusion occurred in five eyes (3.6%). Endophthalmitis (odds ratio (OR), 15.49; 95% confidence interval (CI), 1.70 to 2038.56; *p* = 0.010), endogenous endophthalmitis (OR, 18.73; 95% CI, 3.22 to 125.21, *p* = 0.002), orbital cellulitis (OR, 320.54; 95% CI, 29.67 to 44801.64; *p* < 0.001), implant size (OR, 0.50; 95% CI, 0.30 to 0.79; *p* = 0.004), and hydroxyapatite for the implant (OR, 0.07; 95% CI, 0.00 to 0.66; *p* = 0.016) were risk factors for implant extrusion in univariate logistic regression analysis. Multiple logistic regression analysis identified orbital cellulitis as the only risk factor for extrusion (OR, 52.98; 95% CI, 2.18 to 15367.34; *p* = 0.009). Conclusions: Evisceration with primary orbital implantation is a feasible option in endophthalmitis, but the risk of extrusion should be taken into consideration. When performing evisceration in a patient with orbital cellulitis, secondary implantation should be carried out only after any infection is controlled.

## 1. Introduction

Evisceration is a surgical procedure in which the intraocular contents are removed while the sclera, Tenon’s capsule, conjunctiva, and optic nerve are preserved [1]. Evisceration has relative advantages over enucleation, including improved postoperative fornices and implant motility, easier prosthetic fitting, and generally improved cosmesis [2,3].

Although evisceration was first introduced as a way to prevent the intracranial spread of infection in the presence of severe endophthalmitis [4] researchers have yet to reach a consensus about whether to conduct primary orbital implantation in the presence of endophthalmitis, due to the increased risk of implant extrusion [5,6,7,8]. In the past, many practitioners have favored secondary orbital implantation in endophthalmitis [5,6]. Several authors have recently reported that the increased risk of implant extrusion is insignificant, thus evisceration with primary orbital implant in endophthalmitis has been viewed as an acceptable option [7,8].

However, implant extrusion that occurs after primary implantation is a severe complication that causes difficulties for both the surgeon and the patient. From our clinical experience, endophthalmitis appears to be associated with an increased likelihood of extrusion after primary implantation, but to date research on the risk factors of implant extrusion has been insufficient.

Therefore, we analyzed the risk factors for extrusion of orbital implants after evisceration to explore whether evisceration with primary implantation is a reasonable option in various situations.

## 2. Methods

### 2.1. Subjects

This retrospective study was approved by the Institutional Review Board of the Chungnam National University Hospital, and adhered to the tenets of the Declaration of Helsinki. All methods were carried out in accordance with relevant guidelines and regulations. We retrospectively reviewed the medical records of patients who underwent evisceration with primary implant placement by a single surgeon (SBL) from January 2005 to December 2019 at Chungnam National University Hospital. Age, sex, underlying systemic diseases, axial length of the fellow eye, the cause of evisceration, endophthalmitis type, implant type and size, and preoperative computed tomography (CT) findings were evaluated. All patients were treated using the same evisceration techniques, intraoperative cultures and smears, suture materials, postoperative wound care regimen, and follow-up schedule. All patients were followed up for at least 6 months after evisceration to detect extrusion after surgery. A waiver of informed consent was approved by the Ethics Committee of Chungnam National University Hospital due to the retrospective nature of the study.

Patients were divided into three groups based on the cause of evisceration: endogenous endophthalmitis, endophthalmitis without endogenous origin, and non-endophthalmitis causes. Patients diagnosed with systemic disease (e.g., liver abscess) were recruited for the endogenous endophthalmitis group; only unilateral eyes (the eye operated earlier) were included, even if both eyes were affected. The endophthalmitis without endogenous origin group included patients with endophthalmitis after trauma, including corneal ulcer and intraocular surgery. The non-endophthalmitis group included visually impaired patients experiencing pain, such as those with phthisis bulbi or glaucoma. Patients with an existing or suspected ocular, orbital, lid, paranasal sinus, or cranial malignancy, as well as those with concurrent facio-orbital fractures or immunosuppression were excluded. Facial CT enhancing images were collected the day before each patient’s surgery, and patients were divided into three groups according to their CT findings: those without cellulitis, those with preseptal cellulitis, and those with orbital cellulitis.

Four implant types were used in this study: hydroxyapatite, porous polyethylene, porous silicone, and non-porous silicone. The type and price of the implant were fully explained to the patient and then decided upon consultation.

### 2.2. Surgical Technique

Surgery was performed under general anesthesia and on intravenous antibiotics. A 360° peritomy was performed with Westcott scissors. Tissue handling with toothed forceps was minimized. An incision was made at the limbus with a sharp blade, and the cornea was removed with Stevens scissors. A dialysis spatula was used to dissect the uveal tissues from the sclera. The uveal contents were removed with an evisceration spoon until all of the pigment was scraped from the scleral pocket. Any remaining uveal tissues were denatured with absolute alcohol, followed by vigorous cleansing and copious irrigation. Radial relaxing incisions were made in the anterior sclera between the extraocular muscle. A 360° posterior sclerotomy was performed, rounding behind the equator to enlarge the scleral pocket. The implant size was estimated using a sizer. After the appropriately sized implant was inserted, the scleral shell was closed with interrupted 5-0 Vicryl (Ethicon, Somerville, NJ, USA) sutures. The posterior tenon was closed with interrupted 5-0 Vicryl sutures, the anterior tenon with interrupted 6-0 Vicryl (Ethicon Inc., Somerville, NJ, USA) sutures, and the conjunctiva with running 6-0 Vicryl sutures. An appropriately sized conformer was inserted and ofloxacin ointment was applied. The pressure dressing was kept in place for 4 to 6 days.

### 2.3. Statistical Analyses

Statistical analyses were computed using statistical package R (version 3.5.0, R Foundation for Statistical Computing, Vienna, Austria) and SPSS statistical software for Windows (version 21.0, IBM Corp., Armonk, NY, USA). Continuous variables are presented as the mean ± standard deviation. Probability values of *p* < 0.05 were considered statistically significant. Data were analyzed using independent *t*-test and Fisher’s exact test for the baseline demographics. Univariate and multivariate logistic regression analyses were performed to evaluate the factors associated with implant exposure. Using standard maximum likelihood logistic regression with highly unbalanced dependent variables can underestimate the probability and bias standard errors. The analysis of rare events, such as implant exposure in this study, requires penalized likelihood models. Firth logistic regression (FLR) is one such method for logistic regression. It uses Firth’s bias reduction, an ideal way to handle separation in logistic regression to reduce bias. Therefore, FLRs were conducted in R using the “logistf” package, with the results interpreted in the same manner as traditional logistic regressions. 

## 3. Results

### 3.1. Demographics

In total, 140 patients who underwent evisceration were evaluated (Table 1). There were 92 men and 48 women, and the mean age was 58.6 ± 17.1 years (range: 17–88 years). Extrusion occurred in 5 of the 140 eyes (3.6%). No statistically significant differences were observed in age, sex, laterality, underlying systemic diseases, or axial length of the fellow eye between the implant extrusion and the no implant extrusion groups (Table 1).

The average size of the implant was 20.2 ± 1.5 mm in the group without extrusion and 18.0 ± 2.5 mm in the group with extrusion, a statistically significant difference (*p* = 0.001, Table 2).

The 140 cases of evisceration included 12 patients with endogenous endophthalmitis, 49 patients with endophthalmitis with non-endogenous origin (corneal ulcer, perforation with trauma, or postoperative endophthalmitis), and 79 patients without endophthalmitis (phthisis bulbi or glaucoma). Three of the five extruded patients had endogenous endophthalmitis and two were patients with endophthalmitis with non-endogenous origin. There was no extrusion in the non-endophthalmitis group, and 25.0% (3/12) and 4.1% (2/49) experienced extrusion in the endogenous endophthalmitis group and the non-endogenous endophthalmitis group, respectively.

Before evisceration, all patients were evaluated for the presence of preseptal cellulitis or orbital cellulitis through facial CT imaging. Of the 140 patients, 15 had preseptal cellulitis and 9 had orbital cellulitis. No extrusion was observed in patients without cellulitis or with preseptal cellulitis. Of the nine patients with orbital cellulitis, five had implant extrusion (55.6%) (Figure 1).

The most frequently used implant was hydroxyapatite (75 cases), followed by non-porous silicone (37 cases), porous polyethylene (18 cases), and porous silicone (10 cases). Implant extrusion occurred in three cases with non-porous silicone and in one case each with porous polyethylene and porous silicone implants.

### 3.2. Risk Factors for Orbital Implant Extrusion after Evisceration

Endophthalmitis (odds ratio (OR), 15.49; 95% confidence interval (CI), 1.70 to 2038.56; *p* = 0.010), endogenous endophthalmitis (OR, 18.73; 95% CI, 3.22 to 125.21; *p* = 0.002), and orbital cellulitis in CT findings (OR, 320.54; 95% CI, 29.67 to 44,801.64; *p* < 0.001) were risk factors for implant extrusion, based on univariate logistic regression analysis results (Table 3). Implant size (OR, 0.5; 95% CI, 0.30 to 0.79; *p* = 0.004) and hydroxyapatite (OR, 0.07; 95% CI, 0.00 to 0.66; *p* = 0.016) were negative risk factors for implant extrusion. Age, sex, systemic diseases, and axial length of the fellow eye were not significant risk factors for extrusion. Multiple logistic regression analysis revealed that orbital cellulitis was the only risk factor for extrusion (OR, 52.98; 95% CI, 2.18 to 15,367.34; *p* = 0.009).

## 4. Discussion

Over the years, indications for evisceration have expanded to include both infectious and noninfectious intraocular inflammation, resulting in a total loss of vision, end-stage glaucoma, and post-traumatic severe ocular injuries [9]. Compared with enucleation, evisceration is easier, quicker, and less disruptive to tissues, therefore leading to fewer complications and better ocular motility and eventual cosmesis [10,11]. It also protects against the theoretical risk of meningitis after enucleation of an infected eye [12]. Evisceration includes placing an orbital implant within the scleral shell for improved postoperative cosmetic rehabilitation of the socket [13,14]; however, this procedure has been associated with many complications, including decreased motility, cosmetically unacceptable enophthalmos (due to inadequate volume replacement), infection, implant migration, exposure, and extrusion [9]. Above all, implant extrusion is one of the frequent and severe complications [1,13,15].

Implant extrusion following evisceration was first noted in 1939 when Burch [3] reported a 25% extrusion rate. Researchers have attempted to identify risk factors for implant extrusion after evisceration, of which endophthalmitis was considered the most dominant [2,5,6]. As a consequence, many have advocated for no primary implant in cases of endophthalmitis and have argued that an implant should only be inserted as a secondary procedure, months or years after evisceration [2,6]. Some surgeons prefer not to perform evisceration until the ocular infection is eradicated [5]. Eliminating infection before implant insertion has led to delayed primary wound closure in patients with an infected wound [12,16]. Although delayed primary closure has certain theoretical advantages, primary implantation of orbital implants avoids prolonged hospitalization and the need for two surgeries [8]. For this reason, some in the field prefer to take the small risk of implant extrusion, as opposed to not placing an implant or subjecting the patient to a secondary procedure [7,17]. Studies of primary implant placement in infected eyes undergoing evisceration report low complication rates and good implant retention [12,18,19,20,21]. Practice patterns have recently shifted toward the use of primary orbital implants in these patients [22].

In the current study, no implant extrusion was observed in the non-endophthalmitis group (Table 2) [18], similar to the outcomes reported in previous studies [17,22]; thus, there appears to be no problem with the surgical technique itself. The implant extrusion rate in the endophthalmitis group was 8.2%, and the difference in implant extrusion rates between the two study groups (endophthalmitis versus non-endophthalmitis) was statistically significant. Thus, endophthalmitis is a potential risk factor for implant extrusion, which is consistent with previous studies [5,6,12,16], There is still a debate as to whether primary or secondary implantation is better in patients with endophthalmitis. The primary implant technique is simple, more cost-effective, and more convenient for patients [18]. However, if implant extrusion occurs, it will result in additional surgeries and more pain for the patient, at a greater cost than the planned secondary implantation [6]. To our knowledge, no previous studies have attempted to identify risk factors for extrusion through univariate and multivariate analyses. Gupta et al. [23] attempted to analyze risk factors for implant extrusion using univariate analysis, but their results were inconclusive. Here, we identified the risks through univariate analysis of hypothetical risk factors, on which multivariate analysis was performed to consider the interactions among factors. Using univariate logistic regression analysis, endophthalmitis (OR, 15.49; 95% CI, 1.7 to 2038.56; *p* = 0.010) and endogenous endophthalmitis (OR, 18.73; 95% CI, 3.22 to 125.21; *p* = 0.002) were risk factors for implant extrusion (Table 3). However, when analyzed in multivariate analysis, neither were critical risk factors for implant extrusion. Many authorities believe that eyes infected with a virulent protease-secreting organism, such as Pseudomonas, are at higher risk of implant extrusion [19,24,25]. However, it is difficult to use this information in clinical practice. It takes a long time to detect causative organisms before surgery, and it is possible that the bacteria cannot be detected. Instead, preoperative CT findings can easily and quickly predict the probability of extrusion after evisceration with primary implantation. In our study, orbital cellulitis in the CT results (OR, 320.54; 95% CI, 29.67 to 44,801.64; *p* < 0.001) was a risk factor for implant extrusion in univariate logistic regression analysis, and multiple logistic regression analysis revealed that orbital cellulitis was the only risk factor for extrusion (OR, 52.98; 95% CI, 2.18 to 15,367.34; *p* = 0.009) (Table 3). There is also debate regarding whether the implant type affects extrusion [17,26]. Some studies have reported that porous implants cause fewer extrusions [26], while others have reported the opposite [17]. In our study, there was no significant difference in the extrusion rate between porous and non-porous groups. Our comparison of the rate of extrusion according to implant type revealed that hydroxyapatite (OR, 0.07; 95% CI, 0.00 to 0.66, *p* = 0.016) was a negative risk factor for implant extrusion in the univariate logistic regression analysis (Table 3). On the other hand, the non-porous silicone implant had a borderline significance with an OR of 4.14. We speculate that the result of low extrusion risk in patients using hydroxyapatite implants demonstrated selection bias. When the conjunctival of the Tenon’s capsule is severely inflamed, wound healing may be impaired and increase the likelihood of extrusion. Although there is a debate about endophthalmitis as a risk factor for extrusion, inexpensive non-porous silicone implants were actively recommended for patients with endophthalmitis. After non-porous silicone implants became no longer available in Korea, inexpensive types of implant among the porous implants were recommended and frequently used, cutting down the usage of the relatively expensive hydroxyapatite implants. If non-porous silicone implants were still available, it is highly expected that only non-porous silicone implants would be used in all five exposed cases. In this case, the OR of the non-porous silicone implant would have been higher and statistically significant. Therefore, a randomized prospective study is necessary to evaluate the risk rate according to implant type without selection bias. Karesh and Dresner [26] argued that implant extrusion is related to the placement of an oversized implant. In contrast, Liu et al. [19] reported that a smaller implant did not necessarily involve a lower extrusion risk. In the current study, the axial length of the fellow eye did not differ between extrusion and non-extrusion groups. However, the size of the implant was statistically significantly smaller in the group with extrusion (*p* = 0.001). Using univariate logistic regression analysis, implant size (OR, 0.50; 95% CI, 0.30 to 0.79, *p* = 0.004) was a negative risk factor for implant extrusion, but it was not a risk factor using multivariate analysis. Implant size also appears to be affected by selection bias, similar to implant type. In general, for optimal cosmetic results, the largest implant is selected without creating undue tension in the wound closure. However, for patients who appear to be at high risk of extrusion during surgery, there is the possibility that the surgeon would choose a smaller-sized implant than usual to reduce the risk.

This study is the first to apply multivariate analysis to determine whether endophthalmitis is a risk factor for extrusion. However, it had several limitations. First, the patients were distributed over a 15-year period. This should not have a significant impact on the results, as the patients in this study underwent surgery by a single surgeon and there was no significant change in the surgical technique during this period. Second, despite being a risk factor, the numbers of patients with endogenous endophthalmitis and orbital cellulitis were relatively small. Recruiting more patients and additional analyses would yield statistically much stronger results. Lastly, it is likely that there was a selection bias in the results. Our univariate analysis results indicated that size and type of implant are risk factors for extrusion, although they were not identified as risk factors using multivariate analysis. As previously mentioned, we cannot ignore the tendency that the operator tended to choose smaller and cheaper implants for patients who seem to be at high risk of implant exposure. Multivariate analysis showed that this was not a decisive factor, but a prospective, randomized study may clarify the influence of the implant on extrusion.

## 5. Conclusions

In conclusion, evisceration with primary orbital implantation is a feasible option in endophthalmitis; however, the risk of extrusion should be taken into consideration. In particular, when performing evisceration in a patient with orbital cellulitis, it is desirable to perform secondary implantation only after any infection is controlled.

## Figures and Tables

**Figure 1 jcm-10-03329-f001:**
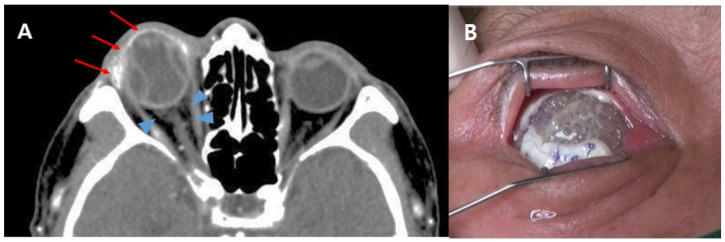
(**A**) Preoperative facial computed tomography (CT) image of a patient with implant extrusion. An axial postcontrast CT image shows enhancement on the anterior globe and lacrimal gland (arrows) and haziness in the retrobulbar area (arrowheads). Endophthalmitis with orbital cellulitis was diagnosed based on CT findings. (**B**) At 2 weeks follow-up after evisceration, the implant was exposed through a dehiscent wound and the implant was extruded on the next day.

**Table 1 jcm-10-03329-t001:** Demographic features of patients.

	No Extrusion (*n* = 135)	Extrusion (*n* = 5)	*p* Value
Age (years, mean ± SD)	58.1 ± 17.3	71.4 ± 15.9	0.093 *
Sex (male, %)	89 (65.9)	3 (60.0)	0.999 ^†^
Laterality (right, %)	73 (54.1)	4 (80.0)	0.379 ^†^
HTN (%)	31 (23.0)	1 (20.0)	0.999 ^†^
DM (%)	32 (23.7)	1 (20.0)	0.999 ^†^
AXL of fellow eye (mm, mean ± SD)	23.1 ± 2.7	23.8 ± 1.5	0.546 *

HTN: hypertension; DM: diabetes mellitus; AXL: axial length. * Independent t test. ^†^ Fisher’s exact test.

**Table 2 jcm-10-03329-t002:** Distribution of risk factors and extrusion rate.

	No Extrusion (*n* = 135)	Extrusion (*n* = 5)	Extrusion Rate (%)	*p* Value
Implant size (mm, mean ± SD)	20.2 ± 1.5	18.0 ± 2.5		0.001 *
16 mm (*n*, %)	5 (3.7)	2 (40.0)	28.6	0.020 ^†^
18 mm (*n*, %)	13 (9.6)	2 (40.0)	13.3	0.089 ^†^
20 mm (*n*, %)	78 (57.8)	0 (0.0)	0.0	0.016 ^†^
22 mm (*n*, %)	39 (28.9)	1 (20.0)	2.5	0.999 ^†^
Cause of evisceration (*n*, %)				
Non-endophthalmitis	79 (58.5)	0 (0.0)	0.0	0.014 ^†^
Phthisis bulbi	58 (42.9)	0 (0.0)	0.0	0.076 ^†^
Intractable glaucoma	21 (15.6)	0 (0.0)	0.0	0.999 ^†^
Endophthalmitis	56 (41.5)	5 (100.0)	8.2	0.014 ^†^
Non-endogenous	47 (34.8)	2 (40.0)	4.1	0.999 ^†^
Endogenous	9 (6.7)	3 (60.0)	25.0	0.004 ^†^
Preoperative CT finding (*n*, %)				
No cellulitis	116 (85.9)	0 (0.0)	0.0	0.999 ^†^
Preseptal cellulitis	15 (11.1)	0 (0.0)	0.0	0.999 ^†^
Orbital cellulitis	4 (3.0)	5 (100.0)	55.6	<0.001 ^†^
Implant type (*n*, %)				
Non-porous silicone	34 (25.2)	3 (60.0)	8.1	0.115 ^†^
Porous	101 (74.8)	2 (40.0)	1.9	0.115 ^†^
Hydroxyapatite	75 (55.6)	0 (0.0)	0.0	0.020 ^†^
Porous polyethylene	17 (12.6)	1 (20.0)	5.5	0.503 ^†^
Porous silicone	9 (6.7)	1 (20.0)	10.0	0.314 ^†^

CT: computed tomography. * Independent t test. ^†^ Fisher’s exact test.

**Table 3 jcm-10-03329-t003:** Risk factors for orbital implant extrusion after evisceration.

	Univariate		Multivariate	
	OR (95% CI)	*p* Value *	OR (95% CI)	*p* Value *
Age	1.06 (0.99 to 1.17)	0.089		
Sex	1.38 (0.22 to 7.32)	0.710		
HTN	1.11 (0.11 to 0.16)	0.918		
DM	0.60 (0.06 to 3.35)	0.583		
AXL of fellow eye	1.05 (0.79 to 1.75)	0.812		
Implant size	0.50 (0.30 to 0.79)	0.004	1.20 (0.62 to 2.44)	0.586
Cause of evisceration				
Endophthalmitis	15.49 (1.70 to 2038.56)	0.010	2.86 (0.01 to 671.83)	0.625
Non-endogenous	1.34 (0.21 to 7.10)	0.738		
Endogenous	18.73 (3.22 to 125.21)	0.002	0.57 (0.03 to 6.49)	0.653
Preoperative CT finding				
Preseptal cellulitis	0.44 (0.22 to 13.20)	0.435		
Orbital cellulitis	320.54 (29.67 to 44,801.64)	<0.001	52.98 (2.18 to 15,367.34)	0.009
Implant type				
Non-porous silicone	4.14 (0.77 to 25.53)	0.096		
Hydroxyapatite	0.07 (0.00 to 0.66)	0.016	0.63 (0.00 to 108.85)	0.812
Porous polyethylene	2.25 (0.22 to 13.20)	0.435		
Porous silicone	4.44 (0.41 to 27.39)	0.184		

HTN: hypertension; DM: diabetes mellitus; AXL: axial length; CT: computed tomography; OR: odds ratio; CI: confidence interval. * Logistic regression analysis.

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
