# Peer review of "Risk Factors for Orbital Implant Extrusion after Evisceration"

_jcm, 2021, doi:10.3390/jcm10153329_

Round 1
Reviewer 1 Report
The manuscript is well written and presented, although it does not provide new insight in the subject.
However, it is interesting to underline the reality of risk of implant extrusion in case of endophthalmitis and/or orbital cellulitis.
Lines 266-267 in the discussion need to be explained, or they should be deleted.
Author Response
Dear Reviewer,
We deeply appreciate your valuable comments and your helpful suggestions about our study. We totally agree with your comments that line 266-267 would be better explained. We previously explained that selection bias occurred with implant size and material, but it would be better to briefly explain once more, so we added some explain as follow:
(Line 267-269) Lastly, it is likely that there was a selection bias in the results. Our univariate analysis results indicated that size and type of implant are risk factors for extrusion, although they were not identified as risk factors using multivariate analysis. As previously mentioned, we cannot ignore the tendency that the operator tended to choose smaller and cheaper implants for patients who seem to be at high risk of implant extrusion. Multivariate analysis showed that this was not a decisive factor, but a prospective, randomized study may clarify the influence of the implant on extrusion
In addition, we accepted the opinion that the English word spell check and format review were necessary, and after requesting a re-examination, some corrections and refinements were made. We are very sorry that we can not describe all the fine spell correction here, but you can see it in the new version of manuscript. Thank you for your deep consideration to improve the quality of our study.
If you have any further suggestions for changes, please let us know. Thank you.
Sincerely yours,
Sung Bok Lee, MD, PhD
Reviewer 2 Report
Well done. Enjoyed reading and feel the results contribute to our knowledge of extrusions.
Author Response
Dear Reviewer,
We deeply appreciate your valuable comments and your helpful suggestions about our study. we accepted the opinion that the English word spell check and format review were necessary, and after requesting a re-examination, some corrections and refinements were made. We are very sorry that we can not describe all the fine spell correction here, but you can see it in the new version of manuscript. Thank you for your deep consideration to improve the quality of our study. If you have any further suggestions for changes, please let us know. Thank you.
Sincerely yours,
Sung Bok Lee, MD, PhD
Reviewer 3 Report
Excellent study; the statistical analysis is outstanding
Author Response

(The authors gave the same response as above.)

Reviewer 4 Report
This is a retrospective cohort study examining risk factors for orbital implant extrusion after evisceration. The article is well written and well reasoned. Despite a small sample size of patients who experienced implant extrusion, the authors explain limitations to the study in a very thorough manner at the end of the discussion section. In the methods section, the authors should clarify how they selected which eye to include if the disease process was bilateral - was the eye with worse vision included, etc.?
Author Response
Dear Reviewer,
We deeply appreciate your valuable comments and your helpful suggestions about our study. We totally agree with your comments that authors should clarify how they selected which eye to include if the disease process was bilateral, in the methods section. Of 140 patients, only one patient had surgery on both eyes. We selected the eye operated earlier and indicated this in the text as follow:
(Line 69- 70) Patients diagnosed with systemic disease (e.g., liver abscess) were recruited for the endogenous endophthalmitis group; only unilateral eyes (the eye operated earlier) were included, even if both eyes were affected.
In addition, we accepted the opinion that the English word spell check and format review were necessary, and after requesting a re-examination, some corrections and refinements were made. We are very sorry that we can not describe all the fine spell correction here, but you can see it in the new version of manuscript. Thank you for your deep consideration to improve the quality of our study.
If you have any further suggestions for changes, please let us know. Thank you.
Sincerely yours,
Sung Bok Lee, MD, PhD